# Iridium and Ruthenium Modified Polyaniline Polymer Leads to Nanostructured Electrocatalysts with High Performance Regarding Water Splitting

**DOI:** 10.3390/polym13020190

**Published:** 2021-01-07

**Authors:** Razik Djara, Marie-Agnès Lacour, Abdelhafid Merzouki, Julien Cambedouzou, David Cornu, Sophie Tingry, Yaovi Holade

**Affiliations:** 1Laboratoire de Physico-Chimie des Hauts Polymères (LPCHP), Université Ferhat Abbas, Sétif 19000, Algeria; djararazik@univ-setif.dz (R.D.); hafid_merzouki@univ-setif.dz (A.M.); 2Institut Européen des Membranes, IEM UMR 5635, University of Montpellier, ENSCM, CNRS, 34090 Montpellier, France; julien.cambedouzou@enscm.fr (J.C.); david.cornu@enscm.fr (D.C.); sophie.tingry@umontpellier.fr (S.T.); 3ChemLab, CEDEX 5, 34296 Montpellier, France; marie-agnes.lacour@enscm.fr

**Keywords:** conducting polymer, polyaniline, electrocatalysis, hydrogen evolution reaction, oxygen evolution reaction, water splitting

## Abstract

The breakthrough in water electrolysis technology for the sustainable production of H_2_, considered as a future fuel, is currently hampered by the development of tough electrocatalytic materials. We report a new strategy of fabricating conducting polymer-derived nanostructured materials to accelerate the electrocatalytic hydrogen evolution reaction (HER), oxygen evolution reaction (OER), and water splitting. Extended physical (XRD, scanning electron microscopy (SEM), energy-dispersive X-ray spectroscopy (EDX)) and electrochemical (cyclic voltammetry (CV), linear sweep voltammetry (LSV), electrochemical impedance spectroscopy (EIS)) methods were merged to precisely characterize the as-synthesized iridium and ruthenium modified polyaniline (PANI) materials and interrogate their efficiency. The presence of Ir(+III) cations during polymerization leads to the formation of Ir metal nanoparticles, while Ru(+III) induces the formation of RuO_2_ oxide nanoparticles by thermal treatment; they are therefore methods for the on-demand production of oxide or metal nanostructured electrocatalysts. The findings from using 0.5 M H_2_SO_4_ highlight an ultrafast electrochemical kinetic of the material PANI-Ir for HER (36 − 0 = 36 mV overpotential to reach 10 mA cm^−2^ at 21 mV dec^−1^), and of PANI-Ru for OER (1.47 − 1.23 = 240 mV overpotential to reach 10 mA cm^−2^ at 47 mV dec^−1^), resulting in an efficient water splitting exactly at its thermoneutral cell voltage of 1.45 V, and satisfactory durability (96 h).

## 1. Introduction

The designing principles of electrode materials for electrochemical or electrocatalytic applications involve a host and guest approach between active sites (where the reaction takes place) and the support (electrons conduction from or to substrate) linked by a contact (mechanical strength and electronic transfer). Chung et al. [1] have shown that an electrode’s electrical resistance (thus its conductivity) strongly affects the metrics of electrochemical activity and this becomes a concern for high mass loadings (increased thicknesses), before concluding that the electrical conductivity should be considered as one of the essential criteria when designing active materials for real electrochemical energy storage/conversion applications. The interest towards conducting polymer (nano)structures is growing yearly in (bio)electrochemistry as either precursors of freestanding electrocatalysts [2,3] or a supports [4,5,6] for active sites based on enzyme/abiotic catalysts. For both purposes, recent studies show that the combination of polyaniline (a conducting polymer) and metallic species (Ni, Co, Mo) produces nanostructured materials with high performance regarding the electrocatalytic hydrogen evolution reaction (HER), oxygen evolution reaction (OER), and water splitting [2,3,7,8].

In heterogeneous electrocatalysis of water splitting, however, metal oxides (MO_2_, M = Ir, Ru) or oxy-hydroxides (MO(OH), M = Ni, Fe, Co, etc.) are the state-of-the-art for OERs [9,10,11], while metals (Rh, Pt, Ir) are suitable for HERs [12,13] (also non-noble [2,14,15,16]), regulated by the oxophilicity character of the surfaces [17,18]. Basically, those metal nanoparticles are initially prepared before the used of carbon black to lower the metal content, which does not enable the maximization of the electrocatalytic performance and minimization of the loss of active sites during the long-term operation. Hence, chemical polymerization in the presence of metallic cations combined with a proper thermal treatment seems to be an elegant approach to develop high-performance electrocatalysts for the half-cell reactions of water splitting (HER and OER), and previous results with polyaniline (PANI) testify to this [3,19,20]. However, given that HER and OER require either metallic or oxidized surfaces, the remaining major challenge is to find a unified method of synthesis that would allow the obtention of either metallic or oxidic nanostructured materials by adjusting the chemical content of the synthesis. To address this challenge, we report in this contribution a strategy based on the polymerization of aniline in the presence of Ru(III) and Ir(III) to synthesize PANI-derived RuO_2_ or Ir nanoparticles. This way, we were able to produce on-demand oxide or metal nanostructured electrocatalysts to trigger the water splitting exactly at its thermoneutral cell voltage of 1.45 V at 25 °C in 0.5 M H_2_SO_4_ with satisfactory durability (96 h). In the half-cell, these electrode materials demonstrated outstanding performance in 0.5 M H_2_SO_4_ towards HER (PANI-Ir-calcined, 36 mV_RHE_ at 10 mA cm^−2^, 21 mV dec^−1^), and OER (PANI-Ru-calcined, 1.47 V_RHE_ at 10 mA cm^−2^, 47 mV dec^−1^).

## 2. Experimental

### 2.1. Materials and Chemicals

Ruthenium (III) chloride hydrate (RuCl_3_·xH_2_O, Premion^®^, 99.99%, Alfer Aesar, Haverhill, MA, USA), iridium (III) chloride hydrate (IrCl_3_·xH_2_O, 99.8%, Alfer Aesar, Haverhill, MA, USA), aniline (ANI, 100%, Alfa Aesar, Haverhill, MA, USA), sulfuric acid (H_2_SO_4_, 98%, Sigma Aldrich, St. Louis, MO, USA), hydrochloric acid (HCl, 37%, VWR, Radnor, PA, USA), ammonium persulfate ((NH_4_)_2_S_2_O_8_, APS, 98%, Merck, New York, NO, USA), isopropanol (iPrOH, 99.5%, Sigma Aldrich, St. Louis, MO, USA), and Nafion^®^ suspension (5 wt%, Sigma Aldrich, St. Louis, MO, USA) were used as-received. A carbon paper electrode (AvCarb MGL190, 190 μm thickness) was obtained from Fuel Cell Earth LL (Stoneham, MA, USA). Water was produced from a Milli-Q Millipore source (New York, NY, USA) (18.2 MΩ cm at 20 °C).

### 2.2. Synthesis of Polyaniline-Based Ruthenium and Iridium Materials

The different materials were synthesized by modifying our initial method [2,3]. Typically, 25 mL solution made of 0.5 M HCl and 0.4 M ANI was first prepared and put into a reactor at 5 °C. Then, another 25 mL solution containing 0.5 M HCl, 0.2 M APS and suitable metal precursor was prepared. Control synthesis was achieved without any addition of metallic species and is referred to as PANI. In total, 547 mg of RuCl_3_·xH_2_O was used for the material PANI-Ru, and 415 mg of IrCl_3_·xH_2_O for PANI-Ir. After adding the second solution to the first one at 5 mL min^−1^, the chemical polymerization proceeded for 13 h. Afterwards, the solvent was eliminated by rotavap and the solid product was dried in an oven at 80 °C overnight. The materials without any calcination were referred to as PANI-Ru-nc and PANI-Ir-nc. The calcination program under air was a heating at 2° C min^−1^ up to 250, 350, and 400 °C, 1 h dwell. The obtained materials are hereinafter termed PANI-Ir-c, and PANI-Ru-c.

### 2.3. Physicochemical Characterization

Power diffraction X-ray (XRD) analysis was performed on a PANalytical Xpert-PRO diffractometer (Malvern Panalytical, Almelo, the Netherlands) (40 kV, 20 mA) equipped with a copper anode at λ(CuKα) = 1.54 Å, and in Bragg-Brentano mode (with 2θ = 20–80°). Scanning electron microscopy (SEM) (Tokyo, Japan), and energy-dispersive X-ray spectroscopy (EDX) (Oberkochen, Germany) analyses were carried out on Hitachi S-4800 FEG, and ZEISS EVOHD 15 microscopes, respectively.

### 2.4. Electrochemical and Electrocatalytic Measurements

For half-cell measurements in a three-electrode setup at 25 °C, the supporting substrate was a rotating disc electrode (RDE, 5 mm) that was polished with the alumina slurry and cleaned in a water ultrasonic bath. Then, a volume of 4 µL of the prepared catalytic ink (ultrasonic mixing of 260 µL water, 100 µL iPrOH, 50 µL Nafion^®^ suspension, and 4 mg material) was drop-casted, and dried at room temperature, resulting in a loading of 0.2 mg cm^−2^. The counter electrode was a large surface area glassy carbon plate. The reference electrode was Ag/AgCl/KCl (3 M) and was isolated from the 0.5 M H_2_SO_4_ electrolytic solution by a Haber–Luggin capillary tip. For standardization, a calibrating curve obtained in H_2_-saturated electrolyte allowed converting the potentials vs. reversible hydrogen electrode (RHE) by the relationship E(V vs. RHE) = E(V vs. Ag/AgCl/KCl (3 M)) + 0.23. Potentiostatic electrochemical impedance spectroscopy (EIS) performed at given potential (see below) and 10 mV amplitude permitted the iR-drop correction. The solution resistance was 2.1–2.4 Ω. For full-cell experiments in a single compartment cell at room temperature (18 ± 2°C), the cut carbon paper electrode (4 cm high, 4 cm wide, and 0.190 mm thick) was washed with iPrOH under shaking and was dried in an oven. Next, a volume of the catalytic ink of either 41 or 82 µL was drop-casted onto each face to reach 0.1 or 0.2 mg cm^−2^. The used potentiostat was SP-150 (Biologic Science Instruments) (Seyssinet-Pariset, France). The accelerated ageing tests were performed by chronopotentiometry at |j| = 10 mA cm^−2^ in the half cell and I = 100 mA in the full cell.

## 3. Results and Discussion

### 3.1. Physicochemical Characterization of the Materials

#### 3.1.1. XRD Analysis

The pristine polymer-based materials were synthesized via the chemical oxidative polymerization of aniline into polyaniline (PANI) in hydrochloric acid by ammonium persulfate. We previously showed that using ammonium persulfate as the oxidizing agent produces an emeraldine form of PANI (with polaron/bipolaron amount of 43% and base amount of 54% [21]) by using a set of physicochemical and electrochemical screenings [2,3,21]. So, the present work focuses on the polymer modified by iridium and ruthenium (Figure 1a) in an attempt to produce hybrid nanostructured materials with high-performance regarding HER and OER after a calcination step.

In Figure 1b, the numerous diffraction peaks in the XRD patterns of the uncalcined materials (PANI-Ir-nc, PANI-Ru-nc) originate from the polymer itself and the crystallized Ir(+III) and Ru(+III) salts. It is important to clarify that a calcination step was introduced to consolidate the material before its use in electrochemistry, during which metal cations might be lost from the surface of the electrode. For PANI-Ru-c, the broad peaks at 27.9°, 35.1°, 39.8°, and 54.2° correspond to the hexagonal structure of the RuO_2_ oxide phase (JCPDS 21-1172) for (110), (101), (200), and (211), respectively. For PANI-Ir-c, the broad peaks at 40.6°, 47.4°, and 69.1° correspond to the cubic structure of metal phase Ir (JCPDS 87-0715) for (111), (200), and (220), respectively. Interestingly, the calcination program under air of 2 °C min^−1^ up to the different dwells (250, 350, and 400 °C) for 1 h was thought to produce RuO_2_ and IrO_2_ materials from chloride salts [22,23]. Our XRD results corroborate well in the case of ruthenium, but invalidate the case of iridium. Indeed, peaks of IrO_2_ were expected to be at 28.0°, 34.7°, 40.0°, and 54.0° (JCPDS 15-870) for (110), (101), (200), and (211), respectively. However, an earlier report of Mamaca et al. [24] on the thermal decomposition of polymeric precursors (the so-called Pechini–Adams method) showed that ruthenium leads to RuO_2_ while iridium produces a mixture of IrO_2_ and Ir, yet they used a pure oxygen atmosphere with a calcination program of 1 °C min^−1^ increasing from room temperature to 250 °C, then 10 °C min^−1^ to 350 °C, and 30 °C min^−1^ to 400 °C (1 h dwell) [24]. It should be pointed out that a high yield of Ir nanoparticles is routinely obtained without any thermal treatment under air (or oxygen) at 250–400 °C [11], and this applies to the synthesis of metal particles in general. While the nobleness discrepancy (Ir being nobler than Ru) could be a plausible explanation [25]—the facts that air produces IrO_2_ [22] while oxygen produces IrO_2_+Ir [24] stipulate that the availability of oxygen in the calcination oven could not be the determining parameter. Hence, the presence of the polyaniline in the starting material might play an important role in this selective thermal treatment by providing a mixture of oxidative and reductive environments. In-depth investigations are ongoing to elucidate this outcome. Indeed, the XRD of the material PANI-Ir-nc suggests that metallic species of iridium already formed after polymerization, but the fact that these particles do not oxidize at the above-mentioned temperatures calls for extensive studies beyond the scope of the present study. A protection by a superficial layer that would prevent oxygen from entering deeply can be postulated. Furthermore, the average crystallite size calculated by the Scherrer’s law is 9.3 nm for PANI-Ir-c (and thus Ir), and 5.6 nm for PANI-Ru-c (and thus RuO_2_).

#### 3.1.2. SEM Analysis

We next utilized electron microscopy for an overview analysis. We specifically combined normal SEM (Figure 2a,c,e,g), and backscattered SEM (Figure 2b,d,f,h) for a better visualization. Images of PANI-Ir-nc (Figure 2b) highlighting the presence of metal particles (white spots) agree with its previous XRD. After calcination, the morphologies of the materials change drastically with a net discrepancy between the metals and nonmetals, as expected because the metals dominate.

#### 3.1.3. EDX Analysis

In order to extract quantitative data from Figure 2, we further utilized EDX. Figure 3a–d display the results obtained for the as-synthesized PANI-based Ir materials before and after calcination. The intensity of the black spots dominant in Figure 3a nearly disappears in Figure 3c, which is in agreement with the EDX spectra (Figure 3b vs. Figure 3d). Figure 3b shows that within the polymerized material PANI-Ir-nc there is unreacted ammonium persulfate ((NH_4_)_2_S_2_O_8_) because of the distribution of 6 at% S and 26 at% O. The amount of 55 at% C and 12 at% N (the extra amount about 1 at% comes from the ammonium persulfate) confirms the expected proportion of 1 atom of N for 6 atoms of C as observed in a previous study on the absence [21] or presence of metallic cations [3], and where the structure of emeraldine form of PANI was established. The low amount of remaining nitrogen suggests that the NH_4_^+^ species was eliminated during the rotavap step, and clearly rule out any complexation of Ir(+III) by NH_3_ ligand. Even though the Cl^−^ initially introduced as HCl was expected to be eliminated during the rotavap step (total molar ratio of Cl/S = 2.9 but experimentally equals to 0.2/5.7 = 0.035), it is worth mentioning that the remaining composition of 0.6 at% Ir and 0.2 at % Cl matches well with the precursor formula. This could indicate that the observed particles are those of IrCl_3_ and/or Ir (in this scenario, Cl comes from emeraldine salt of PANI (polaron/bipolaron amount up to 43% and base [21])). For the calcined material PANI-Ir-c, Cl disappears, indicating that the particles in Figure 3a do not have the same composition as those of Figure 3c. The calcination (Figure 3d) step increases the Ir amount from 7 to 91 wt%. Interestingly, the atomic ratio O/Ir goes from 47.9% to 0.5%, which clearly indicates that IrO_2_ (O/Ir = 2) is not the dominant phase, thus quantitatively substantiating the previous XRD results. The presence of carbon (5 wt%), and oxygen (4 wt%) would suggest that either a carbon and/or iridium-oxide shell surrounds the iridium metal nanoparticles to avoid their deeper nonoxidation at 250–400 °C under air.

Furthermore, the EDX analysis of the ruthenium-based materials (Figure 4a–d) points out radical changes before and after calcination. While the previous results demonstrated that the initially introduced nitrogen in the form of NH_4_^+^ (from the oxidizing agent (NH_4_)_2_S_2_O_8_) can be readily eliminated by the rotavap step to produce PANI-Ru-nc, EDX results in Figure 4b clearly show a high amount of nitrogen as compared to carbon. Supposing that N of the aniline is engaged in the polymerization to produce polyaniline, then 47 at% C would correspond to 8 at% N (1 N for 6 atoms of C). Then, the remaining 15−8 = 7 at% N would suggest that half of 2 at% Ru and 7 at% Cl is used to produce the structure [Ru(NH_3_)_6_]Cl_3_ which is well-known in (bio)electrochemistry as the redox probe [Ru(NH_3_)_6_]^3/2+^ [26,27]. This is true because the distribution of 6 at% S and 24 at% O fits well with that of the unreacted ammonium persulfate ((NH_4_)_2_S_2_O_8_)), which is rational because of the use of an acidic medium that excludes the formation of hydroxides. This complexation could potentially explain the previous difference in the XRDs of PANI-Ir-nc and PANI-Ru-nc (Figure 2b).

After the calcination (Figure 4d), the amount of S decreases from 6 to 3 at% while N disappears completely in the PANI-Ru-c material (at least to a level too low to be detected by EDX). The relatively high amount of carbon (15 wt%) as compared to the case of PANI-Ir-c (5 wt%) could be assigned to the difference in the molecular weight between Ru (102.9 g mol^−1^) and Ir (192.2 g mol^−1^) because the atomic percentage is nearly the same (33–35 at%). For PANI-Ru-c, the oxygen amount is significantly higher (49 ± 7 at%), which leads to an atomic ratio O/Ru = 2.9 ± 0.5. Hence, a chemical composition of RuO_2_ can be postulated owing to the findings from XRD. The extra amount of oxygen could come from the presence of carbon and sulfur that form some oxidized species in air.

#### 3.1.4. EDX Mapping

We next sought to carefully localize and track the positions of the main elements (C, O, S, Ir, Ru) before and after the calcination. To this end, we used EDX mapping and the results are displayed in Figure 5a–d. The first confirmation is that the position of the white spots in the backscattered SEM image of the PANI-Ir-nc material (Figure 5a) matches well with the map of Ir. In addition, the O and Ru signals that do not overlap in the material PANI-Ru-nc (Figure 5b) do so in PANI-Ru-c (Figure 5d), which supports the aforementioned RuO_2_ nanostructure. Conclusively, this set of physicochemical analyses demonstrated our ability to develop a unified method of synthesis that allows the obtention of either metallic or oxidic nanostructured materials by fine-tuning the chemical content of the synthesis. In the next sections, these synthesized PANI-derived RuO_2_ or Ir nanoparticles will be electrochemically characterized before interrogating their electrocatalytic activities towards HER, OER, and water splitting in an acidic medium.

### 3.2. Electrochemical Performance

#### 3.2.1. Electrochemical Characterization

Having demonstrated our ability to manipulate chemical polymerization in the presence of metallic cations to produce on-demand oxide or metal nanostructured materials, we next utilized the method of cyclic voltammetry to electrochemically probe their surfaces. The results are presented in Figure 6. The overlay in Figure 6a clearly reveals that each material responds, in different ways, electrochemically. A deep analysis of the metal-free conducting polymer PANI (Figure 6b) at different scan rates shows the presence of four domains, A to D, corresponding to the characteristic redox processes associated with the well-described oxidation states of PANI [28,29,30]. Specifically, the pair of redox peaks A/C (0.58/0.40 V_RHE_, iR-drop uncorrected) belongs to the transition from leucoemeraldine (fully reduced, oxidation in A) to emeraldine salt (partially oxidized, reduction in C), whereas those of B/D are assigned to the transition from emeraldine salt (reduced, oxidation in B) to pernigraniline (fully oxidized, reduction in D). It was observed that the profile of cyclic voltammetry (CV) of PANI-based materials (no calcination) changes with the number of the cycles, which is well-documented and attributed to an irreversible morphological change of the polymer coupled with the polymer chain restructuring and the mobility of the anions [31,32]. The slopes of 0.71–0.98 in Figure 6c indicate that the adsorption is the limiting process in these electrochemical reactions because a limitation by the diffusion would render a slope of 0.5 [33].

The CVs of the calcined materials were recorded at different scan rates in order to account the impact on both faradaic and capacitive currents. The CV profile of PANI-Ru-c (Figure 6d) without any obvious hydrogen region is characteristic of an electrode made of RuO_2_ material, similar to the behavior of likewise calcination methods [10,22,24]. For PANI-Ir-c (Figure 6e), the behavior is that of Ir metal, similar to other noble metals such as Pt [11,13,34,35]. The low potential region is characterized by the reversible proton adsorption and desorption processes at the metallic surface (IrH_ads_/IrH_des_) between 0.05 and 0.40 V_RHE_. These species will play an important role in the electrocatalytic hydrogen evolution reaction. Then, the double layer region overlaps with the Ir(0) to Ir(III) transition up to 0.8 V_RHE_ before the complex phenomena of Ir(IV)/Ir(III) and Ir(V)/Ir(IV) [11,36]. The electrochemically active surface area determined by using the hydrogen desorption region (after background current correction and using a monolayer charge of 218 µC cm^−2^ [13,34]) was 7 cm^2^ (roughness factor of 35 or specific value of 20 m^2^ g^−1^). The plots of current vs. scan rate at 0.4 V_RHE_ in Figure 6f show that both materials have the same value of capacitance, 9–10 mF. From the average specific capacitance of 35 µF cm^−2^ [37], the determined active surface area is 257–285 cm^2^, which is significantly larger than the value from the previous method. The raison is that the two methods do not probe the same active sites and there is no standard value of the specific capacitance that can range from 11 to 130 µF cm^−2^ [37].

#### 3.2.2. Half-Cell Performance Regarding Hydrogen Evolution Reaction and Oxygen Evolution Reaction

Having electrochemically characterized the as-synthesized materials, we next sought to evaluate their electrocatalytic activity for both hydrogen evolution reaction (HER), and oxygen evolution reaction (OER) by using linear sweep voltammetry (LSV), and chronoamperometry. The obtained results are shown in Figure 7a–f. Basically, the theoretical value is 0 V_RHE_ for HER (H^+^/H_2_ couple), and 1.23 V_RHE_ for OER (O_2_/H_2_O couple), provided that each species of the couple is present (Nernst’s thermodynamics). For HER, the LSV of Figure 7a clearly evidences the best activity of PANI-Ir-c over other materials tested herein or reported with metals [16,38,39,40,41] or without [14,15,16,41,42]. Specifically, the potentials needed to achieve the metric current densities of |j| = 10 mA cm^−2^ for PANI-Ir-c and PANI-Ru-c are −0.036, and −0.196 V_RHE_, respectively. Relative to the literature [14,15,16,38,39,40,41,42], an overpotential of 36 mV is a record value with a loading of only 0.2 mg cm^−2^ while up to 26 mg cm^−2^ is needed for 98 mV [43]. Unlike PANI-Ir-nc, it was observed that PANI-Ru-nc becomes increasingly more active during the cycling, as shown in Figure 7b. These results confirm the previous findings and definitely underpin the conclusion that within the polymerized materials, the metallic cations are present as [Ru(NH_3_)_6_]Cl_3_, and IrCl_3_. Hence, [Ru(NH_3_)_6_]^3−^ is gradually and electrochemically reduced to Ru(0) that next catalyzes HER while a crystallized IrCl_3_ particle cannot be transformed into Ir(0) to catalyze HER. Tafel plots were also created (Figure 7c) to account for the reaction mechanism, as explained below by Equations (1)–(3). It can be concluded that HER (2H^+^ + 2e^−^ → H_2_) is limited by the hydrogen adsorption for PANI-Ru-c (α = 0.78 leading to b = 76 mV dec^−1^) while both hydrogen adsorption and desorption determined the efficiency of PANI-Ir-c (*b* = 21 mV dec^−1^).
(1)$$\mathrm{Volmer}\;\mathrm{step}:\;{H^{+}}_{\left(\mathrm{aq}\right)}+e^{-}\to H_{\left(\mathrm{ads}\right)},\;b=\frac{2.3RT}{\alpha F},\;b=118.2\;\mathrm{mV}\;{\mathrm{dec}}^{-1}\;\mathrm{at}\;25\;{}^{\circ}C$$
(2)$$\mathrm{Heyrovsky}\;\mathrm{step}:\;H_{\left(\mathrm{ads}\right)}+{H^{+}}_{\left(\mathrm{aq}\right)}+e^{-}\to H_{2(g)},\;b=\frac{2.3RT}{(1\;+\;\alpha )F},\;b=39.4\;\mathrm{mV}\;{\mathrm{dec}}^{-1}\;\mathrm{at}\;25\;{}^{\circ}C$$
(3)$$\mathrm{Tafel}\;\mathrm{step}:\;H_{\left(\mathrm{ads}\right)}+H_{\left(\mathrm{ads}\right)}\to H_{2(g)},\;b=\frac{2.3RT}{2F},\;b=\;29.5\;\mathrm{mV}\;{\mathrm{dec}}^{-1}\;\mathrm{at}\;25\;{}^{\circ}C$$
where *b* is the Tafel slope (mV dec^−1^), α is the symmetry coefficient (typically, α = 0.5), F is the Faraday constant (96,485 C mol^−1^), *R* is the ideal gas constant (8.314 J K^−1^ mol^−1^), and *T* is the absolute temperature (273.15 + °C).

Furthermore, the LSV of OER (Figure 7d) shows that for current densities below 50 mA cm^−2^, PANI-Ru-c is the best performing electrode material. Quantitatively, the potential needed to achieve the metric current density of |j| = 10 mA cm^−2^ for PANI-Ir-c, and PANI-Ru-c is 1.525, and 1.472 V_RHE_, respectively. It was expected that the two materials would meet at high current densities as the oxide form MO_2_ (M = Ru, Ir) is the most active for OER. In fact, the metallic iridium in PANI-Ir-c will evolve into oxide (+IV) at high potentials [11,35,36]. For the mechanism, the Tafel slope of 45–47 mV dec^−1^ (Figure 7e) means that the limiting step is the deprotonation of the adsorbed hydroxyl species (from the first step of water splitting) [10,35,44]. Finally, the chronoamperometry tests in Figure 7f conclusively demonstrate that the best pair of materials for the water splitting is PANI-Ru-c as an anode and PANI-Ir-c as a cathode. In such configurations, only a cell voltage of 1.52 V (iR-drop uncorrected) would be needed to achieve the metric current density of 10 mA cm^−2^. For comparison with existing literature, the obtained value of 1.47 V_RHE_ with PANI-Ru-c outperformed other reported Ru/Ir-based electrocatalysts with 1.50–1.65 V_RHE_ in acidic media [10,11,13,23].

#### 3.2.3. Overall Water Splitting

Before entering into discussions, it is important to outline that water electrolysis (splitting) can never start at 1.23 V (at 25 °C), unlike certain misunderstandings based solely on the standard potentials of the two couples H^+^/H_2_ (E° = 0 V_RHE_) and O_2_/H_2_O (E° = 1.23 V_RHE_). In fact, 1.44–1.48 V is the thermoneutral voltage required to take into account the entropy increase (2H_2_O_(l)_ → 2H_2(g)_ + O_2(g)_) [24,45,46].

The efficiency of water electrolysis was evaluated in a simple and single two-electrode cell containing 35 mL of 0.5 M H_2_SO_4_ (the support (blank) was carbon paper electrode of 4 cm high, 4 cm wide, and 0.190 mm thick—i.e., 8 cm^2^ for both external surfaces). The results shown in Figure 8a–d were obtained with PANI-Ru-c as the anode and PANI-Ir-c as the cathode, based on the previous data of Figure 7. Here, we first applied the stepped voltage method to obtain the final polarization curve. Figure 7a shows that the used symmetric approach of ±0.05 V steps (left *Y*-axis) between 1.2 and 2 V (iR-drop uncorrected) leads to a symmetric profile of the current (right *Y*-axis) that quickly stabilizes, as can be observed in Figure 7b for three different trials, thus validating the method. The extracted curves before and after iR-drop correction (Figure 8c) show that the water electrolysis starts at the theoretical thermoneutral voltage of about 1.45 V. The complex-plane Nyquist impedance plots at 1.55 V (Figure 8d) indicate that the deposited electrocatalytic materials do not add any significant ohmic resistance to the electrode. As can be seen in Figure 8c–d, the high ohmic resistance of about 2.3 Ω from the liquid electrolyte and carbon paper electrode are the main cause of the increase in cell voltage. These data obtained in a proof-of-concept test imply a promising result when implemented in practical polymer electrolyte membrane (PEM) water electrolyzers.

We next utilized the method of LSV to determine whether or not the used method influences the recorded data. Figure 9a shows the successive polarization curves recorded at low scan rate of 0.005 V s^−1^ before and after iR-drop correction for the configuration (−)Ir||Ru(+)—i.e., PANI-Ir-c as the cathode and PANI-Ru-c as the anode. The three polarization curves emphasize no meaningful drop in performance during cycling. Additionally, the water electrolysis starts at a cell voltage and achieves a current of 80 mA (roughly 10 mA cm^−2^) at the same respective values as found earlier. Additionally, it can be seen that the control experiment with carbon paper only (Blank) enables us to rule out any significant contribution of the carbon oxidation within this potential window. We next interrogated the nature of the anode or cathode on the performance by running two different combinations, (−)Ir||Ru(+), and (−)Ru||Ir(+). The results in terms of the polarization curves are reported in Figure 7b, while the electrochemical impedance spectroscopy data are depicted in Figure 7c for complex-plane Nyquist impedance plots and Figure 7d for the Bode diagrams. Bode diagrams show the presence of one time constant “RC”, meaning that the raw data can be modeled by the representative equivalent electrical circuit of R_Ω_ + Q_CPE_//R_ct_ (embedded in Figure 7c) in which R_Ω_ represents the uncompensated ohmic resistance, Q_CPE_ is the constant phase element, and R_ct_ is the charge transfer resistance [47,48]. The results show that the configuration (−)Ir||Ru(+) exhibits the best performance with a 0.17 V decrease in cell voltage. This is substantiated by the reduced charge transfer resistance (inversely proportional to the rate constant k°, thus to the exchange current density j°) even at 1.55 V for (−)Ir||Ru(+); 1.60 V was used for (−)Ru||Ir(+). Note that if the cell voltage increases, R_ct_ will corollary decrease.

For implementation in real electrolyzers, the electrode materials must provide good stability in terms of withstanding a given current to provide hydrogen. To probe this, we performed a durability test at an applied current of I = 100 mA and recorded the polarization curves for dwell times of 0, 24, and 96 h (3.2 mL of water consumed). The recorded LSV at 0.005 V s^−1^ at different stages of the durability test is shown in Figure 9e and the corresponding voltage vs. time trend is reported in Figure 9f. The results highlight satisfactory durability with only +1.13 mV h^−1^ voltage increase during the first 24 h to stabilize at the iR-free voltage of 1.6 V. Considering that the electrocatalyst loading was only 0.2 mg cm^−2^, this is an outstanding longevity when compared to a loading of 2.5 mg cm^−2^ (i.e., 12.5 times higher than herein) needed to maintain about 10 mA cm^−2^ at 1.6 V [38]. Considering that the corrosion of the used carbon support can be very important at this current, the future deployment of these electrocatalysts in conditions of PEM electrolyzers [9,11,22,45,46] holds promise.

Although the membrane electrode assembly (MEA) of real electrolyzers will utilize a relatively high loading of electrocatalysts (1–2.5 mg cm^−2^ [9,11,23]), we finally questioned herein whether the loading could affect or not significantly affect the performance in the present proof-of-concept setup. The optimized configuration (−)Ir||Ru(+) was used with two loadings of 0.1 and 0.2 mg cm^−2^. Given that in preliminary tests 0.1 mg cm^−2^ seemed appropriate, we further ran control experiments with single electrocatalyst systems (i.e., (−)Ir||Ir(+) and (−)Ru||Ru(+)) for a better assessment. The findings are summarized in Figure 10a–c. While there is only a difference of ΔE = 0.035 V between the single electrocatalyst systems (Figure 10a), there is about 0.1 V decrease when they are judiciously chosen to assemble a hybrid cell. Deep analysis by EIS in Figure 10b,c shows ohmic resistance decreases of about 0.1 Ω when the loading is doubly diminished. The same profile of complex-plane Nyquist impedance is found and is similar to previous reports [9,45]. With the Bode diagrams that show one time constant, “RC”, the raw data can be modeled by the representative equivalent electrical circuit of R_Ω_ + Q_CPE_//R_ct_. The increase in the charge transfer resistance at 1.55 V (Figure 10b) directly impacts the value of the current in Figure 10a, about 80 mA for 0.1 mg cm^−2^, and 115 mA for 0.2 mg cm^−2^. Overall, the developed materials from this polyaniline-based strategy are promising.

## 4. Conclusions

In this contribution, we report a polymer-based methodology to engineer high-performance nanostructured electrocatalysts for the hydrogen evolution reaction (HER) and oxygen evolution reaction (OER) in an acidic medium. The chemical polymerization of aniline into polyaniline (PANI) in the presence of metallic cations M(+III) (M = Ir, Ru) and the thermal treatment under air atmosphere were both mastered to produce Ir and RuO_2_ nanoparticles with crystallite sizes of 9.3 nm (Ir) and 5.6 nm (RuO_2_) as determined by power diffraction X-ray (XRD). These structures were unambiguously confirmed by complementary methods of scanning electron microscopy (SEM) coupled to energy-dispersive X-ray spectroscopy (EDX), and cyclic voltammetry (CV). Half-cell characterization in 0.5 M H_2_SO_4_ showed that the PANI-Ir material is suitable for HER (36 mV_RHE_ at 10 mA cm^−2^, 21 mV dec^−1^), while PANI-Ru is suitable for OER (1.47 V_RHE_ at 10 mA cm^−2^, 47 mV dec^−1^). This straightforward method allows the production of on-demand oxide or metal nanostructured electrocatalysts to trigger the water splitting exactly at their thermoneutral cell voltage (1.45 V) with satisfactory durability (96 h), thus contributing to advancements in the field of sustainable H_2_ production.

## Figures and Tables

**Figure 1 polymers-13-00190-f001:**
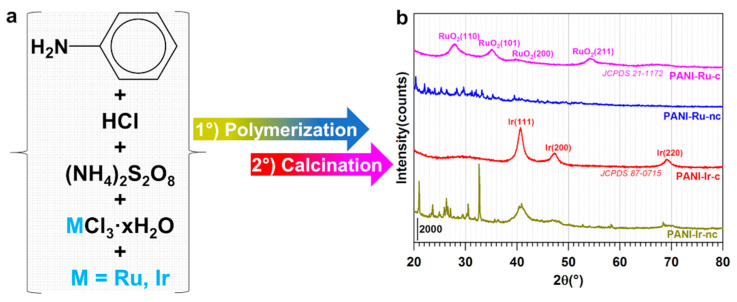
(**a**) Sketch of the developed methodology leading to uncalcined materials (polyaniline (PANI)-Ir-nc, and PANI-Ru-nc), and calcined ones (PANI-Ir-c, and PANI-Ru-c); (**b**) XRD of the prepared materials.

**Figure 2 polymers-13-00190-f002:**
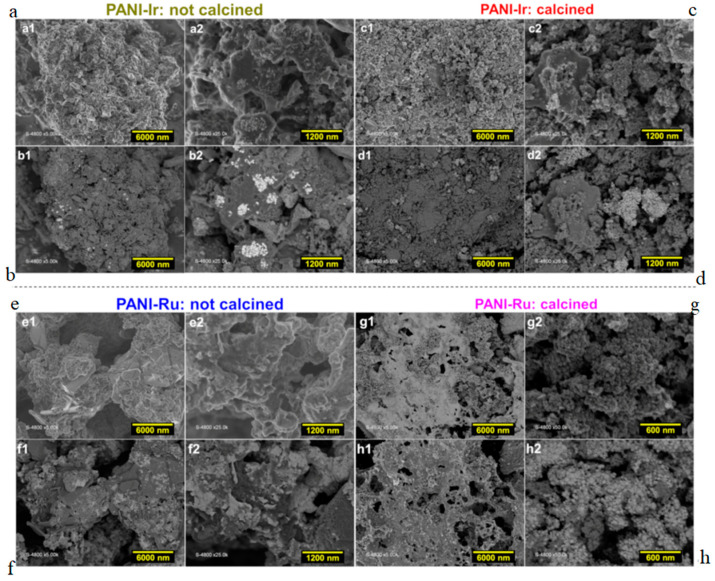
(**a**,**c**,**e**,**g**) Normal scanning electron microscopy (SEM), and (**b**,**d**,**f**,**h**) the corresponding backscattered images of the as-synthesized PANI-based materials: (**a**,**b**) PANI-Ir-nc, (**c**,**d**) PANI-Ir-c, (**e**,**f**) PANI-Ru-nc, and (**g**,**h**) PANI-Ru-c.

**Figure 3 polymers-13-00190-f003:**
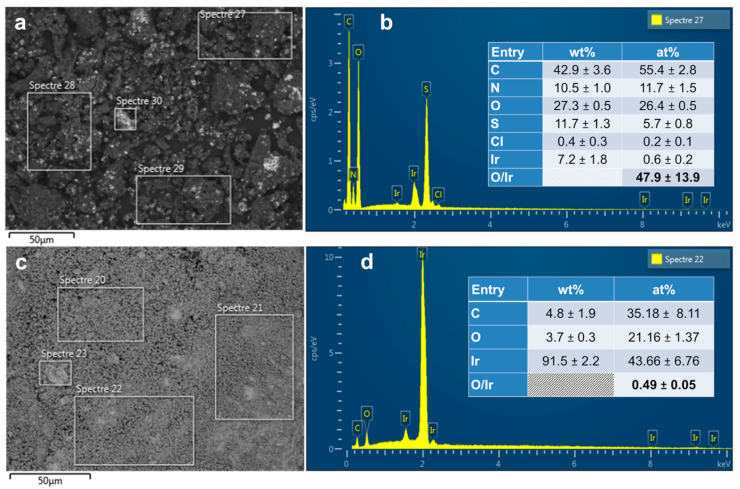
(**a**,**c**) Backscattered SEM image and (**b**,**d**) corresponding energy-dispersive X-ray spectroscopy (EDX) spectra of the as-synthesized PANI-based materials: (**a**,**b**) PANI-Ir-nc, and (**c**,**d**) PANI-Ir-c. Please: Read “Spectre” as “Spectra”.

**Figure 4 polymers-13-00190-f004:**
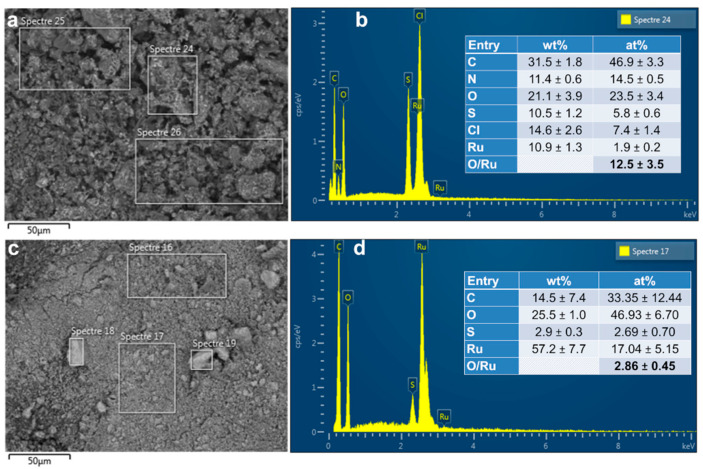
(**a**,**c**) Backscattered SEM image and (**b**,**d**) corresponding EDX spectra of the as-synthesized PANI-based materials: (**a**,**b**) PANI-Ru-nc, and (**c**,**d**) PANI-Ru-c. Please: Read “Spectre” as “Spectra”.

**Figure 5 polymers-13-00190-f005:**
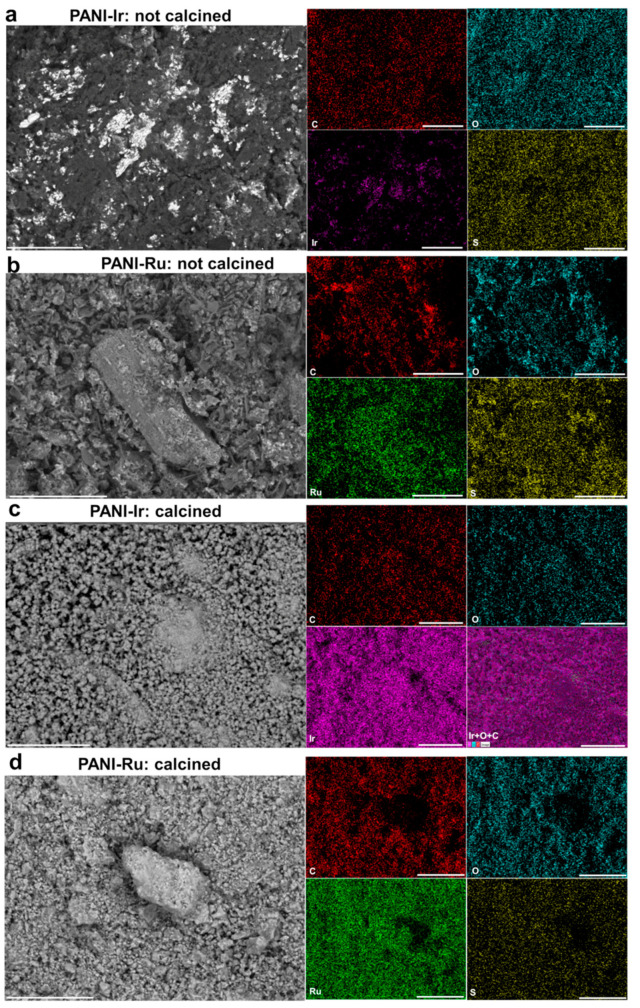
Backscattered SEM images and the corresponding EDX maps of the as-synthesized PANI-based materials: (**a**) PANI-Ir-nc: (**b**) PANI-Ru-nc, (**c**) PANI-Ir-c, and (**d**) PANI-Ru-c.

**Figure 6 polymers-13-00190-f006:**
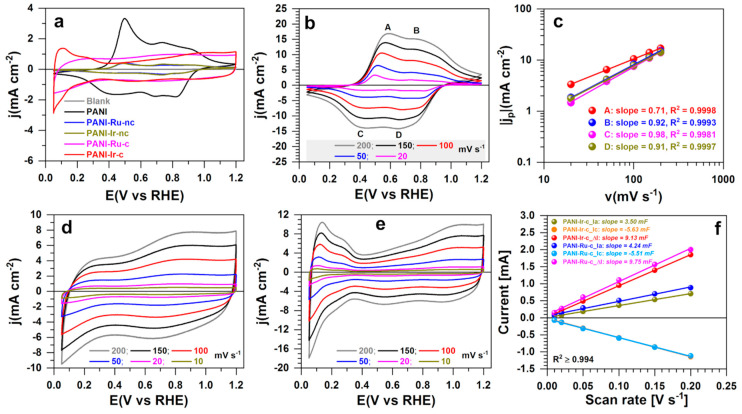
Electrochemical characterization. (**a**) Cyclic voltammetries (CVs) at 20 mV s^−1^ for different electrodes. (**b**) CVs at different scan rates for the PANI material. (**c**) Peak current vs. the scan rate from panel (**b**). (**d**) CVs of PANI-Ru-c at different scan rates. (**e**) CVs of PANI-Ir-c at different scan rates. (**f**) Current vs. scan rate at 0.4 V_RHE_. The support was a rotating disc electrode (RDE) (blank) at 0 rpm, the electrolyte was 0.5 M H_2_SO_4_, the temperature was 25 °C, and the CVs were iR-drop uncorrected.

**Figure 7 polymers-13-00190-f007:**
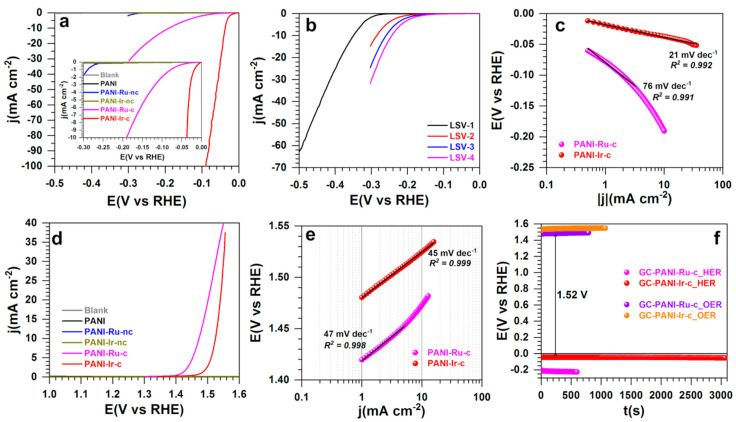
Half-cell hydrogen evolution reaction (HER) and oxygen evolution reaction (OER). (**a**) iR-free linear sweep voltammetry (LSV) of HER at 5 mV s^−1^ for different electrodes. (**b**) iR-drop uncorrected LSV of HER at 5 mV s^−1^ for the PANI-Ru-nc material. (**c**) Tafel plots of HER from panel (**a**). (**d**) iR-free LSV of OER at 5 mV s^−1^ for different electrodes. (**e**) Tafel plots of OER of from panel (**d**). (**f**) Chronopotentiometry curves (iR-drop uncorrected) of HER and OER at an applied current density metric of |j| = 10 mA cm^−2^. The support was RDE (blank) at 1600 rpm, the electrolyte was 0.5 M H_2_SO_4_, and the temperature was 25 °C.

**Figure 8 polymers-13-00190-f008:**
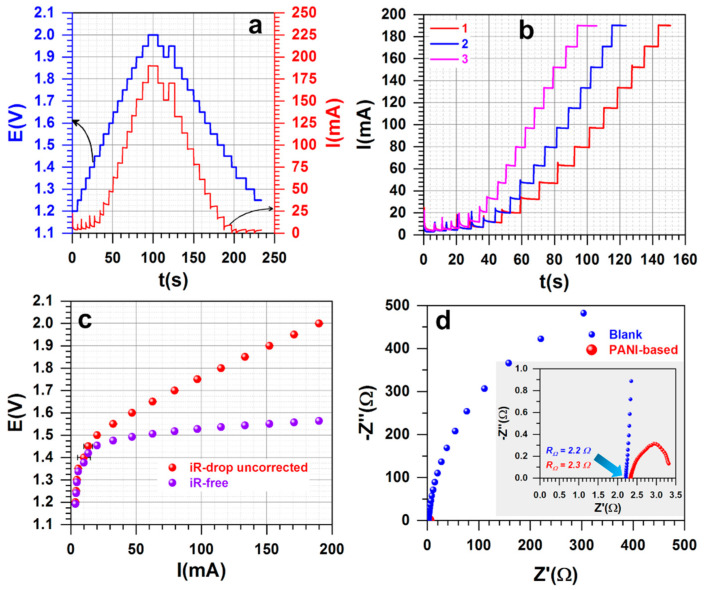
Water splitting realized at room temperature (18 ± 3 °C). (**a**) Illustration of the step-potential method from 1.2 to 2 V and from 2 V to 1.2 V (step of 0.05 V). (**b**) Successive step-potential polarization curves recorded at step of 0.05 V from 1.2 to 2 V. (**c**) Voltage current before and after iR-drop correction (error bars represent one standard deviation, *n* ≥ 3). (**d**) Complex-plane Nyquist impedance plots at 1.55 V. The support (blank) was carbon paper electrode (4 cm high, 4 cm wide, and 0.190 mm thick, i.e., 8 cm^2^ for both external surface), electrolyte was 0.5 M H_2_SO_4_, and cell volume was 35 mL (not stirred).

**Figure 9 polymers-13-00190-f009:**
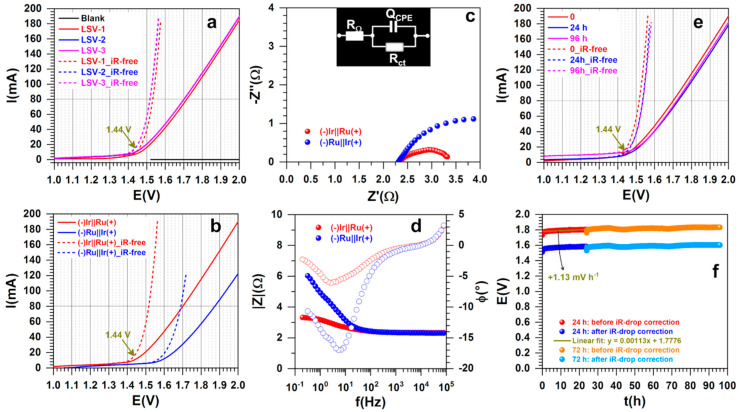
Water splitting realized at room temperature (18 ± 3 °C). (**a**) Successive polarization curves recorded at 0.005 V s^−1^ before and after iR-drop correction for the configuration (−)Ir||Ru(+). (**b**) Polarization curves recorded at 0.005 V s^−1^ before and after iR-drop correction for two configurations. (**c**) Complex-plane Nyquist impedance plots at 1.55 V for (−)Ir||Ru(+) and 1.60 V for (−)Ru||Ir(+): in inset is the the corresponding equivalent electrical circuit. (**d**) Bode diagrams. (**e**) Comparison of the polarization curves recorded at 0.005 V s^−1^ at different stages of the durability test at an applied current of I = 100 mA for 0, 24, and 96 h. (**f**) Corresponding voltage vs. time trend at an applied current of I = 100 mA. “Ir” referred to as “PANI-Ir-c” and “Ru” referred to as “PANI-Ru-c”. The support (blank) was a carbon paper electrode (4 cm high, 4 cm wide, and 0.190 mm thick—i.e., 8 cm^2^ for both external surface), electrolyte was 0.5 M H_2_SO_4_, and cell volume was 35 mL (not stirred).

**Figure 10 polymers-13-00190-f010:**
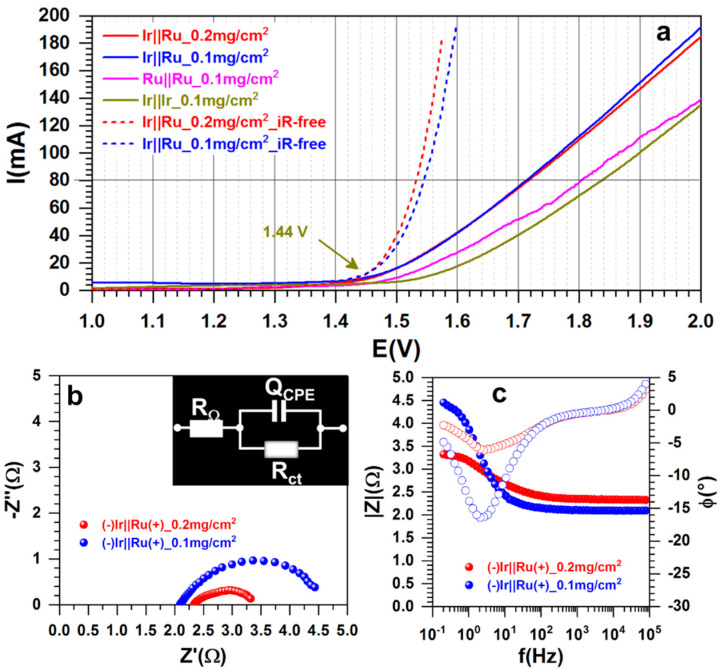
Water splitting realized at room temperature (18 ± 3 °C). (**a**) Polarization curves recorded at 0.005 V s^−1^ before and after iR-drop correction different nature of the electrode material at the anode and cathode. (**b**) Complex-plane Nyquist impedance plots at 1.55 V and in inset the corresponding equivalent electrical circuit is shown. (**c**) Bode diagrams. “Ir” referred to as “PANI-Ir-c”, and “Ru” referred to as “PANI-Ru-c”. The support (blank) was a carbon paper electrode (4 cm high, 4 cm wide, and 0.190 mm thick—i.e., 8 cm^2^ for both external surface), electrolyte was 0.5 M H_2_SO_4_, and cell volume was 35 mL (was not stirred).

## Data Availability

Not applicable.

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
