# Peer review of "Iridium and Ruthenium Modified Polyaniline Polymer Leads to Nanostructured Electrocatalysts with High Performance Regarding Water Splitting"

_polymers, 2021, doi:10.3390/polym13020190_

Round 1
Reviewer 1 Report
Holade et al. submitted their manuscript entitled “Iridium and Ruthenium Modified Polyaniline Polymer Lead to Nanostructured Electrocatalysts with high-Performance towards Water Splitting” for consideration for publication in Polymers. In this study, the authors prepared two polyaniline polymers (PANI-Ir and PANI-Ru) embedded with iridium nanoparticles and ruthenium oxides for hydrogen evolution reaction (HER) and oxygen evolution reaction (OER) respectively. They compared the effects of calcination on these two polymers where the calcinated by a combination of techniques such as X-ray diffraction spectroscopy, energy-dispersive x-ray spectroscopy and cyclic voltammetry. The reviewer is in general pleased by the detailedness and science in this work, which can be accepted for publication, but after solving the following concerns:
- Line 174, the reviewer does not understand how rotavap step can remove NH4+ Ammonium in salt form will have a very high boiling point.
- The practical meaning of the reported electrodes are doubted which is because the use of iridium and ruthenium. They are expensive metals like gold and the electrodes have durability of 96 h only.
- For PANI-Ir-c, the final iridium composition is as high as 91.5wt%. In this case, can one still claim it is “Iridium modified polyaniline polymer”? Also, nitrogen is not detected after calcination, which means the nitrogen in the aniline is gone.
Author Response
Please see the file attachment

Reviewer 2 Report
In this work, the author presented a new strategy based on the polymerization of aniline in the presence of Ru(III) and Ir(III) to synthesize PANI-derived RuO2 or Ir nanoparticles. They have shown the calcination under air is important to produce IrO2 and RuO2 NP selectively. They also used these NPs@PANi for HER, OER, and water splitting reactions. Considering the new design, materials development, perfect material characterization, and activity for HER, OER, and water splitting, I am convening that this work is interesting for publications. However, it will be more interesting if the author provides a more state-of-art study on current HER, OER using metal-, NPs- and metal-free catalyst. A fast checking in google scholar gave me this recent article of OER and HER ACS Appl. Mater. Interfaces 2020, 12, 40, 44689–44699. The author should add this and other related articles for a comparative study. I will be happier if the author provides a surface area study to explore the surface/pore-confinement properties of the polymers. Finally, the author should provide an XPS analysis of Ru and Ir before and after calcination.
With all these additional details, I will happy to recommend this for publications.
Thank you
Author Response
Please see the file attachment
